# The Efficiency of the Removal of Naphthalene from Aqueous Solutions by Different Adsorbents

**DOI:** 10.3390/ijerph17165969

**Published:** 2020-08-17

**Authors:** Alicja Puszkarewicz, Jadwiga Kaleta

**Affiliations:** Department of Water Purification and Protection, The Faculty of Civil and Environment Engineering and Architecture, Rzeszów University of Technology, Al. PowstańcówWarszawy 6, 35-959 Rzeszow, Poland; jkaleta@prz.edu.pl

**Keywords:** naphthalene, removal, adsorption process, activated carbons, clinoptilolite

## Abstract

The paper presents the results of laboratory tests on possibilities to utilize active carbons produced in Poland (AG-5 and DTO) and clinoptilolite for removing naphthalene from a water solution in the adsorption process. The concentration of naphthalene in the model solution was 20 mg/dm^3^. The effects of pH, dose and adsorption time were determined under static conditions. Adsorption kinetics were consistent with the pseudo-second-order model (PSO). Among the applied models, the best fit was obtained using the Langmuir isotherms. The maximum adsorption capacity for the activated carbons (AG-5 and DTO) equaled 24.57 and 30.28 mg/g, respectively. For clinoptilolite, all the analyzed models of adsorption poorly described the adsorption process. The flow conditions were realized by filtration method. On the basis of the obtained results, the breakthrough curves, so-called isoplanes, were prepared and served in turn to determine the adsorption capacities in flow conditions. The total adsorption capacities determined under dynamic conditions of the AG-5 and DTO activated carbons were 85.63 and 94.54 mg/g, respectively, and only 2.72 mg/g for clinoptilolite. The exit curves (isoplanes) were also utilized to determine the mass penetration zone (the adsorption front height), as well as to calculate the rate of mass-exchange zone advance.

## 1. Introduction

Polycyclic aromatic hydrocarbons (PAHs), which include naphthalene, are a group of compounds containing from two to several or even a dozen or so aromatic rings per molecule. In 1987, the International Agency for Research on Cancer (IARC) recognized aromatic hydrocarbons with more than three rings as carcinogenic and mutagenic [1,2].

Polycyclic aromatic hydrocarbons are generally sparingly soluble in water. Their solubility decreases with the increase of their molecular weight. Humic substances and surface-active compounds increase the solubility of PAHs several times [3]. A common form of occurrence of PAHs in water are various types of emulsions [4,5].

PAHs introduced into the water to some degree undergo physical, chemical and biological changes leading to their gradual, though slow, degradation. They are sorbed by suspended matter, aquatic organisms and bottom sediments [6,7]. It was found that the most dangerous PAHs are formed as a product of the metabolism of bacteria, algae and higher plants, hence the natural level of, for example, benzo (a) pyrene is mentioned in the plant world [8].

PAHs present in the environment may be of natural (negligible amounts) and anthropogenic origins. They penetrate into waters with dust pollutants from the air and soil, from which they are washed out by rainwater.

An important source of PAHs is surface runoff from roads. These substances originate from the abrasion of asphalt surface and car tires, as well as from exhaust gases emitted by cars [9,10]. Significant sources of PAHs contaminating surface waters are both municipal and industrial sewage. Various types of failures of oil tankers, as well as leaks from oil pipelines and fuel and oil tanks, are very dangerous [11]. PAHs can get into tap water as a result of leaching from bitumen protective coatings used in water pipelines [6]. The acceptable concentration of PAHs in drinking water according to WHO standards is 0.1 µg/dm^3^ (Regulation of the Minister of Health on the quality of water intended for human consumption, Journal of Laws of 2017, item 2294) [12].

Traditional methods of water treatment, such as coagulation, sedimentation and filtration, remove PAHs (they are sorbed by colloids and suspensions) but not always with satisfactory efficiency [13]. Technological and technical conditions, such as: wastewater quality, flow rate, the use of additional reagents and the problem of formed sediments determine the use of other methods. It is possible to oxidize PAHs with chlorine, but the resulting chlorine products are not neutral for the environment, hence chlorination should not be recommended as a way to remove PAHs from water. Some PAHs (benzo(a)pyrene, antrcen and benzo(a)antrcen) react quickly with chlorine dioxide ClO_2_, and the main products of the reaction are quinones [14].

According to many authors, ozone is safer and the most effective oxidant, which causes almost 100% of their transformations. Naphthalene was successfully oxidized with hydrogen peroxide (H_2_O_2_). The elimination of this compound was 51% [15]. The membrane processes were also used to remove PAHs. The efficiency of their removal from coke wastewater in the ultrafiltration process was 66.6% [16]. The most effective method of PAHs elimination is the adsorption process. Various adsorbents were used, such as activated coke modified with KOH, activated carbons and their modified forms or carbon nanotubes [17,18]. They also conducted the adsorption studies of the naphthalene solution on coal-based activated carbon modified by microwave induction [19]. Granular aluminum-carbon composites removed anthracene (C_14_H_10_) with 99% efficiency [20] and powdered activated carbon adsorbed hexsachlorobenzene from soil with 90% efficiency [21]. The mixture of six PAHs was adsorbed on activated carbon, sand and mineral adsorbent [22]. Additionally tested were unconventional sorbents, e.g., wood ashes roasted at temperatures of 300 °C and 500 °C and modified pine bark that removed phenanthrene and pyrene with 70% efficiency [23,24]. Granular activated carbon also removed pharmaceuticals with high efficiency [25]. Both the sorption capacities and adsorption efficiency depend on many factors; therefore, comparing the obtained results with the literature information is only indicative because the conditions in each study were different.

Natural zeolites, including clinoptilolites, are also effective sorbents. Clinoptilolites are aluminosilicates of skeleton structures containing free spaces filled with large ions and water molecules. The silica content varies in the range of 54.4–66.6% and that of alumina 13.2–18.35%. Clinoptylolites may occur in sedimentary rock (argillaceous schist/shale clay), volcanic rocks (tuffs) and in metamorphic formations. The content of pure clinoptilolite in tuffs is quite considerable and varies in the range of 60–90%. Rich deposits of tuffs are located in Slovakia and Ukraine [7].

Clinoptilolites are biporous substances. They are characterized by primary porosity and secondary porosity. Ion exchange properties were used, among others, for removing different cations of heavy metals [26]. Secondary pores are responsible for the absorption of relatively large particles and have a significant role in many processes and the sorption of organic compounds, e.g., phenol [7].

The purpose of the study was to determine the suitability of activated carbon (AG-5 and DTO carbons with good characteristics and currently produced and available) and clinoptilolite for removing naphthalene from an aqueous solution.

Under static conditions, the adsorption isotherms, pH effect and contact time were determined. Adsorption capacities and filtration isoplanes were determined in dynamic conditions. The introduction of the tested adsorbents as filter beds to the technological systems of water or sewage treatments may contribute to a more effective removal of naphthalene.

## 2. Materials and Methods

### 2.1. Materials

The PAHs adsorption process was performed on the example of naphthalene (C_10_H_8_).

It is often found in industrial wastewater. It is widely used as a raw material and intermediate in the chemical industry for the production of plastics and dyes and, also, as repellent, air freshener and surfactant.

The model solution prepared on distilled water. The naphthalene solution model was pH = 6.8 and concentration of naphthalene Np = 20 mg/dm^3^.

Determination of naphthalene concentration was carried out using an indirect method through absorption measurement (with the wavelength of λ = 254 nm) by Schimadzu UV-1601 Spectrophotometer (Shimadzu Corporate, Tokio, Japan). Quartz cuvettes with 1-cm absorption layers were used. Before carrying out the actual measurements, a model curve was created, and the correlation between naphthalene Np (mg/dm^3^) and absorbency A was determined by the following formula:(1)$$Np=tan32\times A,\;\;tan32=0.624,\;i.e.,\;\;Np=0.624\times A\;\mathrm{mg}/{\mathrm{dm}}^{3}$$
where: tan32—tangent of 32 degrees.

As the basic adsorbents, granular activated carbons manufactured by GRYF-SKAND from Hajnówka: AG-5 and DTO were used (Table 1).

In addition to the activated carbons for the naphthalene adsorption process, clinoptilolite from the town of Niżny Hrabovec near Koszyce (Slovakia) [6], which was only mechanically treated (crushing), was also used. The next step was to isolate the 0.75–1.2-mm fraction, rinse with distilled water and dry at 105 °C (Table 2).

### 2.2. Research under Non-Flow-Through Conditions (Static)

The kinetics of the adsorption process was determined as follows: to 8 bottles with 300 cm^3^ of the model solution of naphthalene each, a fixed dose of the appropriate adsorbent was added (Ag-5 and DTO—0.3 g and clinoptilolite—3 g) and shaken, maintaining different times, from 10 to 180 min. The clarification time was constant and was equal to 30 min. Naphthalene was determined in decanted and filtered solutions. In order to describe the adsorption kinetics, two kinetic models were used most often: the pseudo-first-order (PFO) model defined by the Lagergren equation and the secondary model—pseudo-second-order (PSO)—popularized by Ho et al. [27].

The PFO equation, in a differential form, has the following form:(2)$$\frac{dq\left(t\right)}{dt}=k_{1}\left(q_{e}-q\left(t\right)\right)$$where *t* is the time, *q* the amount of adsorbate bound by the adsorbent (this amount may depend on time), *q_e_* corresponds to the value of *q* in equilibrium, i.e., *q (t → ∞) = q_e_* and *k*_1_ is a constant, called the PFO constant.

During data analysis, the line form is more commonly used:(3)$$\ln(q_{e}-q\left(t\right))=\ln q_{e}-k_{1}t$$

The pseudo-second-order equation (PSO) in a differential form is as follows:(4)$$\frac{dq\left(t\right)}{dt}=k_{2}{\left(q_{e}-q\left(t\right)\right)}^{2}$$where *k*_2_ is a constant.

In order to correlate the experimental data, the linear representation using which of the constant *k*_2_ and *qe* are most often relevant is:(5)$$\frac{t}{q\left(t\right)}=\frac{1}{k_{2}q_{e}^{2}}+\frac{t}{q}$$

The effect of pH on the adsorption process was tested using adsorbents and their doses, such as in the determination of the adsorption kinetics. The time of shaking 2 h and decanting 30 min. The value of pH was changed from 5 to 10.

The effect of the adsorbent dose on the adsorption process: to 8 bottles, 300 cm^3^ of the model solution of naphthalene each and increasing doses of a suitable adsorbent were added. The samples were set at a temperature of 15 ± 2 °C, covered with insulating material and shaken for 2 h. After 30 min of sedimentation, control determination of decanted and infiltrated solutions was carried out.

Isotherms of adsorption were described by models:Freundlich:
(6)qeF=KCe1nF
Langmuir:
(7)$$q_{eL}=q_{m}KC_{e}/\left(1+KC_{e}\right)$$
Dubinin-Radushkevich:
(8)$$logq_{eDR}=-n_{DR}\left(log^{2}\left(KC_{e}\right)\right)+logq_{m}$$
Langmuir-Freundlich:
(9)$$q_{eLF}=q_{m}{\left(KC_{e}\right)}^{n_{LF}}/\left(1+{\left(KC_{e}\right)}^{n_{LF}}\right)$$
where: *q_e_*—adsorption capacity, mg/g, *q_m_*—maximum adsorption capacity, mg/g, *C_e_*—equilibrium concentration, mg/dm^3^, *K*—isotherm constant, which refers to the sorption capacity of the material and *n*—isotherm constant.

Adsorption capacity was determined from the equation:(10)$$q_{r}=\frac{V\left(C_{0}-C_{e}\right)}{m}\;(\mathrm{mg}/g)$$where: *V*—adsorptive volume (dm^3^), *C*_0_
*and C_e_*—initial and equilibrium concentration (mg/dm^3^) and *M*—mass of adsorbent (*g*).

### 2.3. Adsorption Process under Dynamic (Flow-through) Conditions

Dynamic conditions were carried out by the column filtration method. In columns made of organic glass with an inner diameter of 32 mm, a suitable adsorbent was placed so that the filling height was 700 mm. For each adsorbent, its mass (*M*) in the filter bed was precisely determined and were for AG-5—210 g, for DTO—202 g and for clinoptilolite—606 g. The filtrations were carried out from top to bottom. The filter columns were wrapped with “termaflex” insulating foam in order to maintain a constant temperature of 21 °C to protect against light and to prevent the growth of algae.

At the beginning of each filtration cycle, the filtration rates were changed from 5 to 20 m/h, and the optimum speed was set for the given adsorbent.

After each passing of 5 dm^3^ through the bed, samples were taken for the determination of naphthalene. The filtration process each time lasted until the bed was exhausted, i.e., the point where the spill concentration equaled the concentration of the input solution.

On the basis of the results were prepared isoplanes (breakthrough curves), which, after describing with mathematical equations, were used to calculate the sorption capacity (useful PAu and total PAc) sorbents used. The calculations were carried out in accordance with the diagram presented in Figure 1.

Total adsorptive capacity *PAc* (g/kg) of specific material was calculated from the following formula, Equation (11):(11)$$PAc=\frac{Oc}{M}$$

The total amount of naphthalene retained in the column, *Oc*, was calculated from the formula:(12)$$Oc=DFBA-DFB=ABFD-\int_{D}^{F}f\left(x\right)dx$$where *DFBA*—area of the DFBA rectangle, representing the quantity of compounds introduced to the filtration system (the point of bed exhaustion), *DFB*—area under the curve (isoplan), representing the quantity of compounds not retained on the bed (the point of bed exhaustion), M—mass of bed prior to filtration process (*g*), *C*—concentration in the effluent (mg/dm^3^), *C*_0_—initial concentration (mg/dm^3^) and *C_p_*—concentration at the bed breakthrough point (mg/dm^3^).

Usable adsorptive capacity *PAu* (g/kg), was calculated as follows, Equation (13):(13)$$PAu=\frac{Ou}{M}\;(g/\mathrm{kg})$$

The amount of naphthalene retained in the column to the breakthrough point (*Ou*), was calculated from the formula:(14)$$Ou=DGHA-DGI=AGHD-\int_{D}^{G}f\left(x\right)dx$$where *DGHA*—area of the DFBA rectangle, representing the quantity of compounds introduced to the filtration system (the breakthrough point, C = Cp), *DGI*—area under the curve (isoplan), representing the quantity of compounds not retained on the bed (the breakthrough point, C = Cp) and *M*—mass of bed prior to the filtration process (*g*).

The breakthrough curves were used to determine the mass transfer zone, which was calculated using the Michaels and Treybal equation [28]:(15)$$Ho=H\frac{tw-tp}{tw-\left(1-\varphi \right)\left(tw-tp\right)}$$
where *Ho*—height of the adsorption front, cm, *H*—bed height, cm, *tw*—working time of the bed to the point of exhaustion, min, *tp*—working time of the bed up to the break-through point, min, and *φ*—sphericity coefficient of the output curves.

The speed of movement of the mass exchange zone *u* (cm/min) was calculated according to the formula:(16)$$u=\frac{Ho}{tw-tp}$$

## 3. Results and Discussion

### 3.1. The Kinetics of the Adsorption Process

The process of absorption consists of several stages. The first stage is the transport of the adsorbed particle from the solution’s bulk phase to the adsorbent boundary layer, followed by external diffusion (on the adsorbent surface), internal diffusion in the adsorbent structure and the last stage involves the reactions from the adsorbate to the active site. The reaction in the active sites is the fastest, but the adsorption kinetics limit the slowest processes. Under static (nonflow) conditions, these phenomena are diffusion at the interface and inside the pores of the adsorbent. The adsorption in diluted solutions is affected by the time and speed of shaking. The effect of contact (shaking) time on the adsorption process with constant doses of adsorbents is shown in Figure 2. The concentration of naphthalene close to the adsorption equilibrium (adopted as equilibrium) for all were obtained at 2.0 hours.

The modeling process comes down to matching the empirical results with Equations (3) and (5) and choosing the one that better correlates the data. Choosing the right model will not explain the mechanisms controlling the adsorption rate in the system, but it can be helpful in determining the factors limiting the process speed (e.g., pH or temperature and others) [29].

The kinetic curves PFO for the adsorbents tested made in function *f(t) = log(qe – q(t)*) are shown in Figure 3. The graphs for the PSO were prepared in the function *f(t) = t/q(t)* and are shown in Figure 4. Rate constants *k*_1_ (1/h) and *k*_2_ (mg/g·h) were calculated from the inclination and displacement coefficients of the straight lines. All calculated rate constants, standard deviations (*σ*) and determination coefficients R^2^ are shown in Table 3.

The fitting of the pseudo-second-order model to the experimental data indicates that the process was controlled by chemisorption.

### 3.2. The Effect of pH

The effect of pH on the adsorption process with constant doses of adsorbents is presented in Figure 5.

The effectiveness of naphthalene adsorption on activated carbons was higher at pH 5 and 7. In an alkaline environment, the reduction of naphthalene occurred to a lesser extent. The weaker adsorption capacity of the activated carbons observed in an alkaline environment at pH = 8, 9 and 10 was the result of the electrostatic repulsion of negative charges located on the surface of the adsorbents and adsorbate. Similar relationships (a decrease of adsorption with increasing pH) were observed on other active carbons [30,31]. In the case of clinoptilolite, the pH did not have a significant effect on the adsorption process.

### 3.3. The Effect of the Adsorbent Dose

The course of the naphthalene adsorption process using increasing doses of adsorbents is shown in Table 4.

The effectiveness of the adsorption process carried out under static conditions depended on the doses of adsorbents used and increased with their increase. The best efficiency (percentage reduction) was found for DTO I AG-5 carbons at the dose of 3.0 g/dm^3^; it was 95% and 94%, respectively. Clinoptilolite at the dose of 30 g/dm^3^ adsorbed naphthalene only at 63%.

### 3.4. Isotherms of Adsorption

On the basis of the obtained results, adsorption isotherms were prepared; they were described by mathematical equations, constants were determined and the adsorption capacities of the sorbents used were calculated. Constant adsorption isotherms of activated carbons and clinoptilolite in relation to naphthalene are shown in Table 5. The adsorption curves are shown as a function of the amount of naphthalene adsorbed onto the adsorbent at equilibrium versus the naphthalene concentration in aqueous solution (Figure 6, Figure 7 and Figure 8).

As can be seen, the experimentally obtained data for AG-5 and DTO follow all analyzed adsorption isotherms, providing initially a sharp increase of adsorbed naphthalene and tending to approach a maximum adsorption capacity. The low values of both the approximation of the standard deviation (σ) and the mean error (ME) and high determination coefficient (R^2^) show that all applied models well-describe the adsorption process of naphthalene on carbons, but Langmuir’s isotherm was the closest to the experimental conditions. This indicates that the adsorption mechanism proceeded according to the theory of volumetric filling of micropores on a heterogeneous surface of adsorbents.

The maximum adsorption capacity for AG-5 and DTO equaled 24.57 and 30.28 mg/g, respectively. For clinoptilolite, all the analyzed models of adsorption poorly described the adsorption process. The maximum adsorption capacity for Langmuir (biggest R^2^ = 0.732) was 0.897 mg/g.

The adsorption data fitted the Langmuir isotherms, indicating that the adsorption is a monolayer. This suggests that adsorption depends on the availability of active sites on the adsorbent surface.

### 3.5. Test Results under Flow through Conditions

Modeling of the adsorption and desorption processes in dynamic conditions is based on the determination of breakthrough curves [32].

Tests were carried out under dynamic conditions at a speed of 20 m/h (stop time 1.6 min) for AG-5 and DTO and 10 m/h (stop time 3.1 min) for clinoptilolites described by breakthrough curves (Figure 9).

The prepared breakthrough curves are described by the following mathematical equations:

AG-5:(17)$$y=3\cdot 10^{-10}x^{4}-4\cdot 10^{-7}x^{3}+0.0002x^{2}-0.0338x+1.3596;R^{2}\;=\;0.9746$$

DTO:(18)$$y=\;2\cdot 10^{-10}x^{4}-4\cdot 10^{-7}x^{3}+0.0002x^{2}-0.0345x+1.4612;\;R^{2}\;=\;0.9741$$

Clinoptilolite:(19)$$y=0.0006x^{2}-0.0581x+8.808,\;R^{2}\;=\;0.9714.$$

Naphthalene adsorption efficiency for both activated carbons was 100%; for clinoptilolite, it was only 65%. Based on the isoplane equations, the total and useful sorption capacity were calculated, determining the bed breakthrough point at the level of 0 mg/dm^3^ for the activated carbons and at 7 mg/dm^3^ for the clinoptilolite (Table 6).

As it was determined on the basis of tests carried out under static conditions, the best adsorbent was DTO carbon, and the worst was clinoptilolite. Despite their inferior adsorptive characteristics, clinoptilolite may be applied in filtration systems prior to filters with activated carbons.

The adsorption capacity values determined on the basis of isotherms (static conditions) were much smaller than the capacity values calculated on the basis of isoplanes (dynamic conditions). This is in-line with the accepted model of the adsorption process. During tests carried out under static conditions, the adsorption equilibrium was established once and limited further possibilities of the adsorption of pollutants. Adsorption conducted under flow-through conditions can be equated to a process carried out under static conditions, where the adsorbent is added in portions (a stepwise process) and, thus, a better utilization of its adsorptive properties is obtained. Moreover, under dynamic conditions (long life deposits), it is possible to obtain a higher adsorption capacity due to the adsorption of naphthalene on previously adsorbed particles and biodegradation.

From the exit curves (isoplanes), the working time of the bed to the breakthrough point *tp* and to exhaustion *t* was read, and the sphericity coefficient of the output curves was determined *φ*, and then, the height of the *Ho* adsorption front and the velocity of the mass transfer zone *u* were calculated (Table 7).

Adsorbents used under flow-through conditions can also be evaluated on the basis of the height of the mass-exchange front and the speed of the adsorption zone shift. Better adsorbents are those that have a lower mass exchange front and a slower speed of the adsorption zone [33].

The height of the adsorption front for the adsorbents tested in each case was lower than the height of the adsorptive bed layer, which indicates the usefulness of the adsorbents tested in the removal of naphthalene.

The height of the adsorption front and the speed of the mass exchange zone shift for the activated carbons were comparable. Much higher *Ho* and *u* values had a clinoptilolite, which confirms its inferior adsorptive properties compared to naphthalene.

A better active carbon to remove naphthalene was DTO, because it obtained the highest adsorption capacity (also under static conditions). The total adsorption capacities determined under dynamic conditions of the AG-5 and DTO activated carbons were 85.63 and 94.54 mg/g, respectively, and only 2.72 mg/g for clinoptilolite. For comparison, Samiee, L et al. [34] showed that, in studies using mesoporous silica-modified activated carbon, the adsorption capacities for naphthalene were obtained, ranging from 54 mg/g to 161 mg/g depending on the initial concentration of naphthalene, varying from 5 to 30 mg/dm^3^.

For comparison, in another study conducted by Smol [16], a mixture of six PAHs, including naphthalene, was used. The adsorption efficiency depended on the adsorbent used; for active carbon, it was 98%, for sand, 75%, and for mineral sorbent, 22%.

According to other authors, the effectiveness of the process of the adsorption of organic compounds on activated carbons, depending on the type of pollution, is defined at 70–98% [35]. In addition, research conducted by Ho and Newcombe showed that [36], in the long-term of column filtration, on activated carbon, biodegradation may appear next to the adsorption process, which increases the efficiency of adsorption. A completely depleted adsorbent can be regenerated by thermals.

## 4. Conclusions

The total adsorption equilibrium was determined within two hours, but the process occurred most intensively in the first 30 minutes. In general, for all adsorbents, the kinetic parameters (PSO) reflected the trends observed with the experimental data.The adsorption of naphthalene on activated carbons occurred best with pH 5–7. The pH value of the adsorptive solution had little effect on the adsorption process using clinoptilolite.The analysis of isotherm constants showed that the best adsorbent was activated carbon DTO. The Langmuir model best described the experimental data. The maximum adsorption capacity was 30.28 mg/g.Definitely better results were obtained in the column adsorption process. The naphthalene adsorption efficiency for both activated carbons was 100%. The total adsorption capacity for DTO was determined as 94.54 mg/g.Taking the height of the adsorption fronts and the speed of the mass-exchange zone shift and values of the adsorption capacities determined in non-flow-through and flow-through conditions into account, the tested adsorbents can be ranged as follows: DTO > AG-5 > clinoptilolite.

## Figures and Tables

**Figure 1 ijerph-17-05969-f001:**
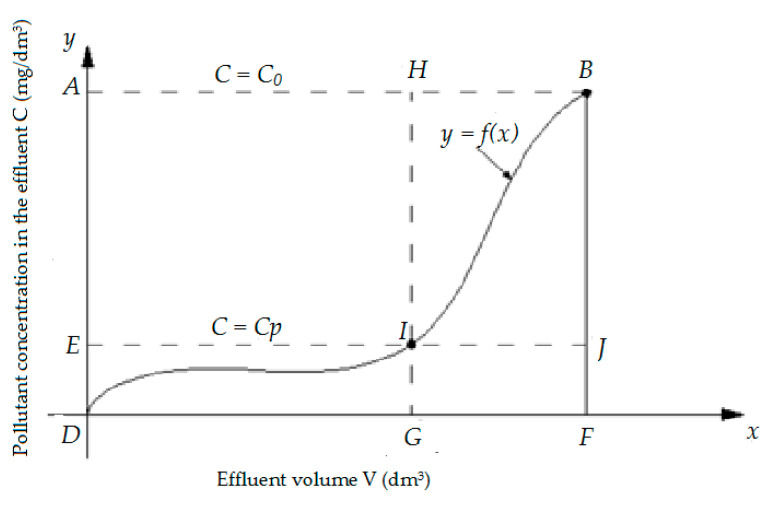
Support drawing for calculation of the total and usable adsorptive capacity.

**Figure 2 ijerph-17-05969-f002:**
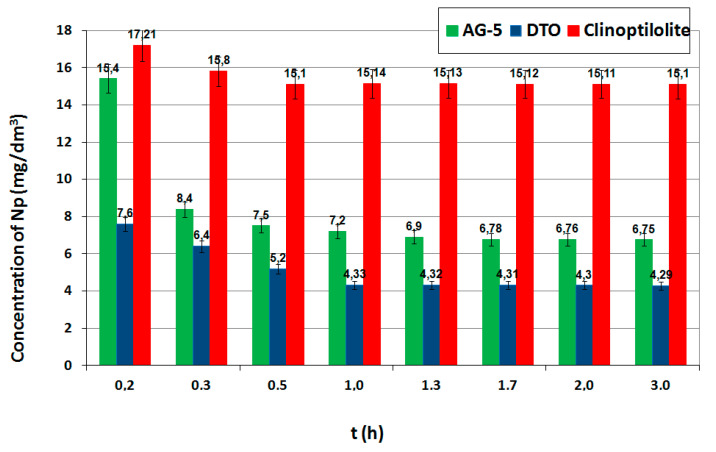
Effect of contact time.

**Figure 3 ijerph-17-05969-f003:**
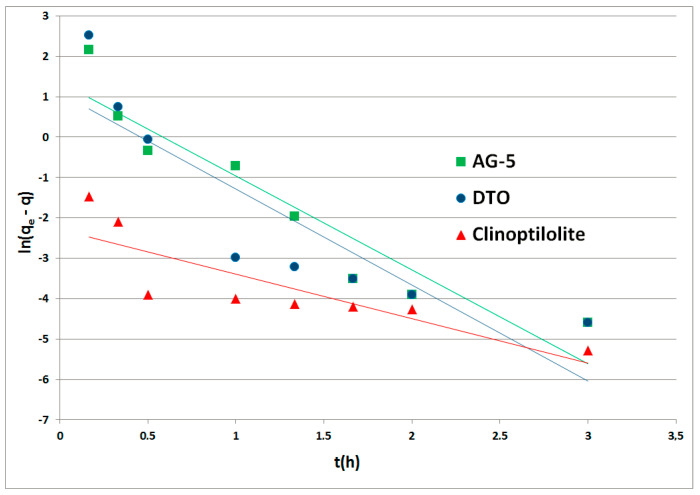
Naphthalene adsorption kinetics of pseudo-first-order.

**Figure 4 ijerph-17-05969-f004:**
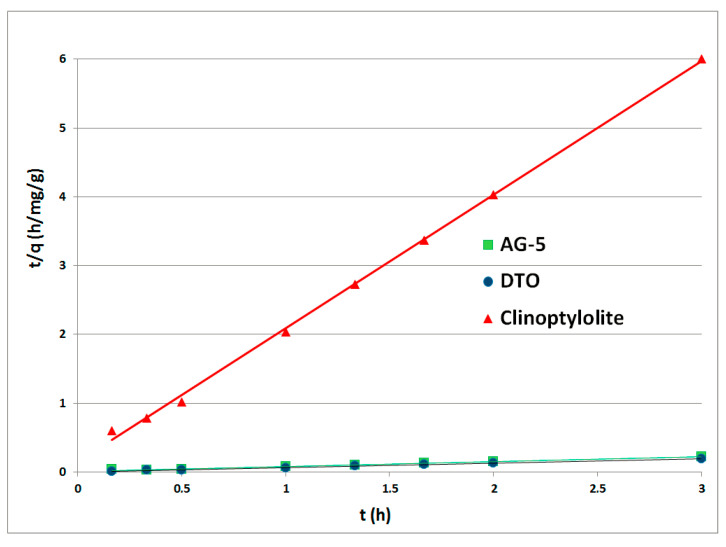
Naphthalene adsorption kinetics of pseudo-second-order.

**Figure 5 ijerph-17-05969-f005:**
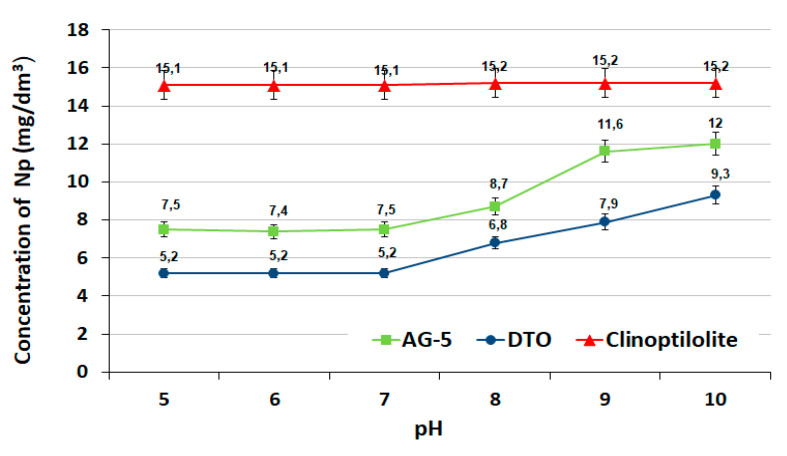
Effect of pH.

**Figure 6 ijerph-17-05969-f006:**
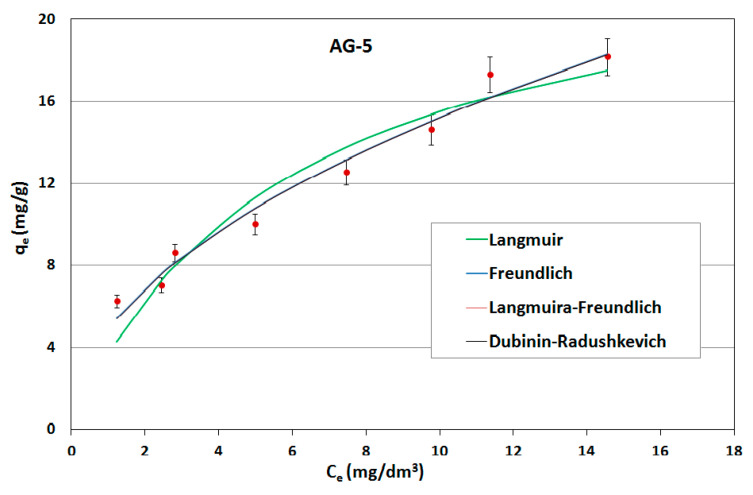
Naphthalene adsorption isotherms for AG-5.

**Figure 7 ijerph-17-05969-f007:**
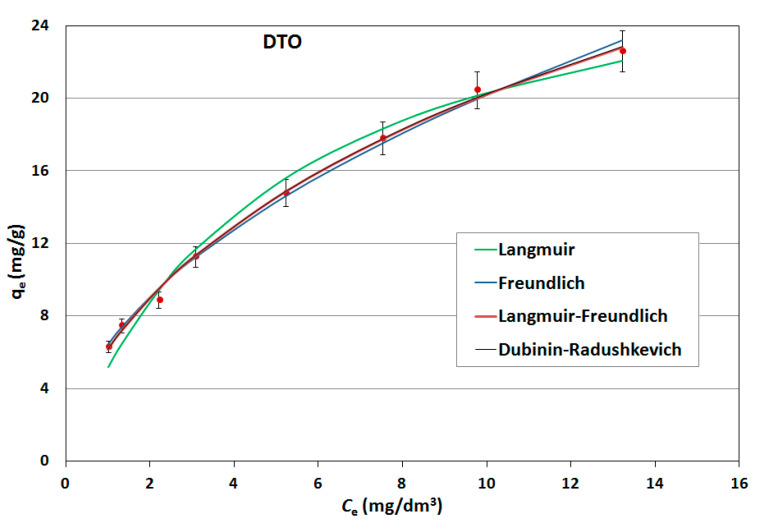
Naphthalene adsorption isotherms for DTO.

**Figure 8 ijerph-17-05969-f008:**
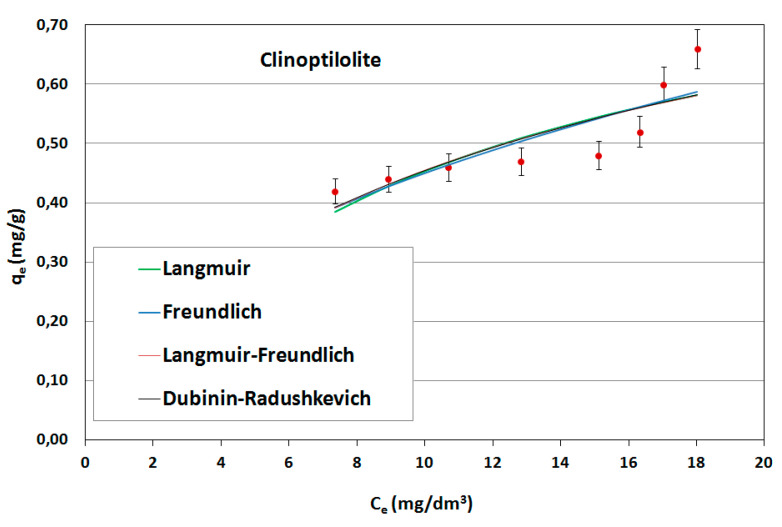
Naphthalene adsorption isotherms for clinoptilolite.

**Figure 9 ijerph-17-05969-f009:**
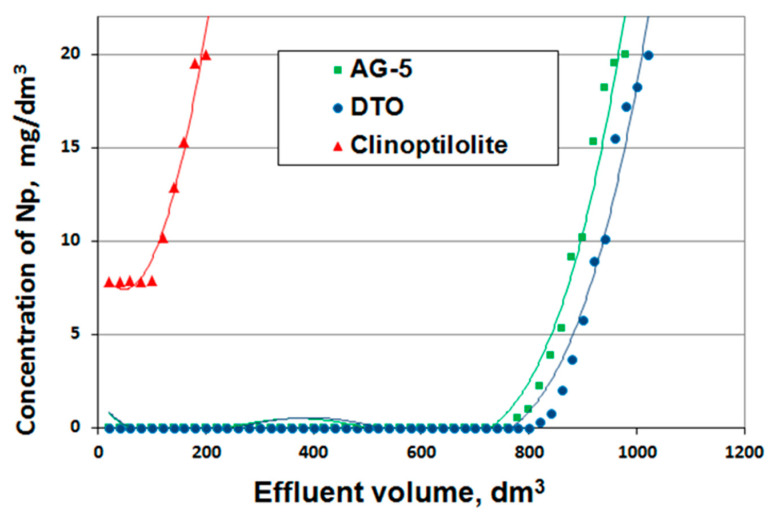
Naphthalene adsorption isoplanes.

**Table 1 ijerph-17-05969-t001:** Characteristics of granular activated carbons (manufacturer’s data).

Designation	Unit	AG-5	DTO
Bulk mass	g/dm^3^	390–410	360–380
Specific surface area	m^2^/g	950–1050	1100–1250
Iodine number	mg/g	900–1000	min. 750
Methylene number	cm^3^	min. 25	min. 50
Granulation	mm	0.75–1.2	0.75–1.2
Total volume of pores	cm^3^/g	0.8–0.9	0.85–1.0
Adsorption of phenol	%	4	4

**Table 2 ijerph-17-05969-t002:** Characteristics of clinoptilolite.

**Physical Properties**	
Hardness according to the Mohs scale	3.5–4
Porosity %	20–30
Impregnability %	30–40
Total volume of pores cm^3^/g	0.08
Specific density g/cm^3^	2.16
Specific surface area m^2^/g	31.8
**The Contents of the Compounds**	
SiO_2_	70–72%
Al_2_O_3_	12–13%
Fe_2_O_3_	1.5–2.0%

**Table 3 ijerph-17-05969-t003:** Rate constants for PFO and PSO.

Adsorbent	Pseudo-First-Order (PFO)			Pseudo-Second-Order (PSO)		
	k_1_ (h^−1^)	R^2^	*σ*	k_2_ (mg/g·h)	R^2^	*σ*
AG-5	2.33	0.8920	2.3632	2.16	0.9911	0.0679
DTO	2.38	0.7803	2.5850	1.71	0.9999	0.0600
Clinoptilolite	1.10	0.7141	1.2505	26.86	0.9988	1.8648

k_1_, k_2_—rate constant for PFO and PSO, respectively; R^2^—determination coefficients; *σ*—standard deviations.

**Table 4 ijerph-17-05969-t004:** Effects of doses of adsorbents used on the removal of naphthalene from a model solution.

AG-5		DTO		Clinoptilolite	
Dose, g/dm^3^	Concentration of Nphthalene, mg/dm^3^	Dose, g/dm^3^	Concentration of Naphthalene, mg/dm^3^	Dose, g/dm^3^	Concentration of Naphthalene, mg/dm^3^
0.3	14.5	0.3	13.2	3	18.0
0.5	11.4	0.5	9.8	5	17.0
0.7	9.8	0.7	7.5	7	16.3
1.0	7.5	1.0	5.2	10	15.1
1.5	4.9	1.5	3.1	15	12.8
2.0	2.8	2.0	2.2	20	10.7
2.5	2.4	2.5	1.3	25	8.9
3.0	1.2	3.0	1.0	30	7.4

**Table 5 ijerph-17-05969-t005:** Values of the constants of the isotherms.

Adsorbent	Constants of Isotherms			ME	*σ*	*R^2^*
	*K*	*n*	*q_m_*			
**Freundlich**						
AG-5	4.866	2.024	-	6.12	0.7774	0.975
DTO	6.394	2.004	-	2.35	0.4366	0.996
Clinoptilolite	0.158	2.208	-	6.78	0.0479	0.716
**Langmuir**						
AG-5	0.170	-	24.57	9.90	1.2718	0.933
DTO	0.203	-	30.28	6.74	0.8357	0.984
Clinoptilolite	0.102	-	0.89	7.44	0.0515	0.732
**Dubinin-Radushkevich**						
AG-5	0.05	0.495	20.52	6.11	0.7794	0.975
DTO	0.028	0.575	48.27	2.26	0.3492	0.997
Clinoptilolite	0.034	0.617	1.05	7.02	0.0498	0.693
**Langmuir-Freundlich**						
AG-5	0.04	0.495	23.26	6.12	0.7790	0.975
DTO	0.029	0.576	47.77	2.26	0.3495	0.997
Clinoptilolite	0.057	0.735	0.95	7.15	0.0506	0.684

ME: mean error; K, n—isotherm constant; *q_m_*—maximum adsorption capacity.

**Table 6 ijerph-17-05969-t006:** Adsorption capacities of carbons and clinoptilolite in relation to naphthalene determined under flow conditions.

Adsorbent	Adsorption Capacities Determined under Flow Conditions, mg/g	
	Useful, PAu	Total, PAc
AG-5	72.38	85.63
DTO	79.21	94.54
Clinoptilolite	2.01	2.72

**Table 7 ijerph-17-05969-t007:** Height of the naphthalene adsorption front on activated carbons and clinoptilolite and the speed of the mass-transfer zone.

The Type of Adsorbent	Coefficient of Sphericity of Output Curves, φ	The Height of the Ho Adsorption Front, cm	The Speed of Movement of the Mass-Exchange Zone u, cm/min.
AG-5	0.696	16.865	0.0112
DTO	0.704	16.128	0.0107
Clinoptilolite	0.353	51.740	0.0758

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
