# Peer review of "The Efficiency of the Removal of Naphthalene from Aqueous Solutions by Different Adsorbents"

_ijerph, 2020, doi:10.3390/ijerph17165969_

Round 1
Reviewer 1 Report
This manuscript presented the efficiency of different adsorbents for naphthalene removal in aqeous solutions. The study is interesting but needs a lot of improvement and additional information.
Comments:
1. What is the novel part of this study? Please state this in the introduction.
2. The background for naphthalene is quite long. Please give an in-depth introduction on your proposed adsorbents (clinoptilolite, active carbon). Recommended References:
-Thermal stability and heavy metal (As5+, Cu2+, Ni2+, Pb2+ and Zn2+) ions uptake of the natural zeolites from the Philippines, 2019, Mater. Res. Express, 6, 085204.
-Adsorption of naphthalene from aqueous solution on coal-based activated carbon modified by microwave induction: microwave power effects, 2015, Chem. Eng. Process. Process Intensif., 91, 67–77.
3. Please fix all the units (supercript), formulae and equations. Also give proper citations for all the formulae, equations and figures (Fig. 1, copyright permission) you used.
4. Why did you choose clinoptilolite as an adsorbent material to be compared with activated carbon?
5. Please give a table comparing the efficiencies of your materials with the existing studies.
6. Please improve all the figures. It's hard to distinguish the differences in some lines (Fig. 5-8).
7. Explain why the Langmuir's isotherm best described the experimental adsorption data.
Author Response
Thank you very much for your review.
Detailed explanations are provided in annex.

Reviewer 2 Report
This paper reported a series of experiments testing the performance of two selected activated carbons for adsorption of naphthalene in water. Some solid and meaningful data were shown and discussed. However, the paper has some problems. Please see below.
General comments:
The authors did not explain why they tested these two activated carbons. It seems that the authors conducted some routine tests to evaluate the performance of these two activated carbons. What information are the authors trying to disseminate? If they intend to show the usefulness of the tested activated carbon, they need to (a) conduct tests on different contaminants, not just limited to naphthalene, and (b) evaluate other adsorbents in parallel trials to show better performance of these two activated carbons.
For all tests, the authors need to conduct replicates and show error bars in figures and standard deviations in tables.
The discussion is not adequate and in-depth. More discussion is needed to explain the mechanism underlying the observations shown in figures and tables.
Specific comments:
Lines 51 to 52: If traditional methods are good, why did you try activated carbon? The traditional methods are cheap.
Lines 72 to 73: You did not clearly indicate how AG-5 and DTO related to other adsorbents you showed in the previous paragraph. If literature has investigated this topic well, why did you do this test? Why did you select these two activated carbons? You need to give explanation to justify your experiments.
Line 77: Why did you test naphthalene, not other PAHs?
Line 81: Please show molar absorption coefficient of naphthalene at 254 nm.
Line 95: Why did you test clinoptilolite? You should give background information and the reason beforehand.
Line 103, “Ag-5 102 and DTO – 1g/dm3, clinoptylolite – 10 g/dm3”: This is confusing. Please indicate the doses by weight.
Line 117: Figure out the mistake with the brackets of the equation.
Line 121: Make this equation in the standard manner as shown above.
Line 152: What is “termaflex” insulation? How did it control algae and maintain the temperature?
Line 158: Table 3 is very simple. Please remove it and show the values in text. Why did you not apply the same mass?
Figure 1: Explain all symbols in the diagram.
Lines 174 to 177, 184 to 187: I do not understand the definitions of PDFBA, PDFB, PDGHA, and PDGI. Please make them clear. What is Cp?
Line 238: Can you prove this through analyses, such as z-potential?
Lines 275 to 276: Can you explain any relationships between adsorption kinetics (shown in Fig 4) and adsorption isotherms (Figs 6 to 8)?
Lines 280 to 281: This is confusing. Give more detailed explanation. Where was coal from?
Lines 300 to 302: Please expand the discussion.
Lines 318 to 319: Explain why this type of activated carbon was better.
Lines 325 to 327: How did the activated carbon perform compared to previous studies? Why did you do this study, if good performance was known?
Author Response
Than you very much for your review.
Detailed explanation is provided in the annex.

Round 2
Reviewer 1 Report
The manuscript is now recommended for publication.
Author Response
Thank you very much for your positive review.
Reviewer 2 Report
I appreciate the authors’ revision. However, I do not think the authors prepared responses carefully. When making responses, the authors need to explain how they made the revision and to specify where the revision was made with line numbers. In the text, please highlight the words that were newly changed. It is very difficult to locate the revised parts.
Response to Line 77: Please add this reason for why you chose naphthalene to the article.
Response to Line 95: You wrote ‘Explanation in the text’. However, I cannot find where you made this explanation. Please specify the revision with line numbers.
Response to Line 152: Add this explanation to the text.
Response to Line 158: Please remove Table 3 and show the data in text.
Response to Fig 1: The symbols in the diagram are not explained yet.
Response to Lines 300 to 302: I can see you added one sentence, but it is still not enough to explicitly explain the mechanism.
Responses to Lines 318 to 319: The current discussion does not deliver an adequate explanation of why this type of activated carbon was better.
The following comments refer to the line numbers in the revised article:
Lines 52 to 53, “…but not always practicable due to many technological and technical limitations”: Please give more details.
Lines 89 to 90: You still did not show clearly why AG-5 and DTO were selected for this study.
Eq. (1): Explain what is tan32.
Eq. (12) to (14): What are DFBA, DFB, DGHA, and DGI? Please explain how you made these short names, i.e., the meaning of each letter.
Author Response
Thank you very much for your comments and directions.
Explanations are in the attachment.
